# Dark Triad Traits and Risky Behaviours: Identifying Risk Profiles from a Person-Centred Approach

**DOI:** 10.3390/ijerph17176194

**Published:** 2020-08-26

**Authors:** Lorena Maneiro, María Patricia Navas, Mitch Van Geel, Olalla Cutrín, Paul Vedder

**Affiliations:** 1Department of Clinical Psychology and Psychobiology, Universidade de Santiago de Compostela, 15782 Santiago de Compostela, Spain; mariapatricia.navas@usc.es (M.P.N.); olalla.cutrin@usc.es (O.C.); 2Institute of Education and Child Studies, Leiden University, 2333 AK Leiden, The Netherlands; mgeel@fsw.leidenuniv.nl (M.V.G.); vedder@fsw.leidenuniv.nl (P.V.); 3Global Center for Applied Health Research, Arizona State University, Phoenix, AZ 85004, USA

**Keywords:** dark triad, Machiavellianism, psychopathy, narcissism, risky behaviour

## Abstract

The relationship between Dark Triad traits and risky behaviours has been shown in recent years. However, few studies have attempted to disentangle this relationship using a person-centred approach. The goal of the current study was to identify subgroups of individuals on the basis of their scores on Machiavellianism, psychopathy, and narcissism and analyse the differences between them in a set of risky behaviours (i.e., frequency of substance use, reactive and proactive aggression, risk perception and risk engagement, and problematic internet use). The sample consisted of 317 undergraduates aged 18–34 (46% males). The results of the latent profile analysis showed five subgroups of individuals that were identified based on their scores on the Dark Triad traits: low-Dark Triad, narcissistic, Machiavellian/narcissistic, psychopathic, and Machiavellian/psychopathic. Overall, the Machiavellian/narcissistic and Machiavellian/psychopathic subgroups showed higher scores for most risky behaviours. The low-Dark Triad scored higher for risk perception. No significant differences between subgroups were found as regards frequency of alcohol, tobacco, and cannabis use. These findings suggest that the combination of the Dark Triad traits lead to more negative outcomes as regards risky behaviour than individual components. Moreover, they highlight the relevance of using a person-centred approach in the study of dark personalities.

## 1. Introduction

The Dark Triad has been conceptualized as a constellation of three socially aversive personality traits at the subclinical level, namely Machiavellianism, psychopathy, and narcissism [1]. Machiavellianism comprises individual features, such as interpersonal manipulation, lack of morality, and instrumental alliance-building [2]; psychopathy refers to a set of interpersonal, affective, and behavioural-aversive traits [3]; and narcissism is characterized by a sense of grandiosity, exploitative entitlement, dominance, and feelings of superiority [4]. The distinctiveness of the Dark Triad traits has been widely investigated [5,6,7,8]; however, much less is known about the individual profiles that underlie the Dark Triad traits. Dark personalities are neither static nor equal in all individuals as some score higher than others in one or more Dark Triad factors, therefore it seems plausible that different subgroups exist that may be differentiated based on their scores on Machiavellianism, psychopathy, and narcissism [9]. 

The study of Dark Triad using a variable-centred approach may lead to inaccurate results because the underlying processes are viewed as universal among all individuals, which may entail, in turn, the underestimation of the heterogeneity of the population [10,11]. By contrast, the person-centred approach considers that individuals can be clustered into homogeneous profiles, which may be the result of a holistic consideration of dark personalities as a dynamic complex system [10,12]. Nevertheless, very few attempts have been made to disentangle the role of dark personalities from a person-centred approach, and existing studies have provided inconsistent results [10,11,12,13]. Given the subclinical nature of dark personalities, the consideration of the Dark Triad traits from a categorical perspective using a person-centred approach might provide valuable insights, both at a theoretical and practical level [13]. The main goal of the current study is, therefore, to identify different individual profiles based on individuals’ scores on the Dark Triad traits and analyse the differences among them in evaluations of risky behaviours. These analyses will allow us to explore the feasibility of using a person-centred approach in the study of the Dark Triad and delve into the differences in risk-taking activities. Consequently, the identification of different risk profiles will contribute to the development of prevention and intervention strategies adapted to the specific needs of individuals [14].

### 1.1. Dark Triad and Risky Behaviour

The intrinsic characteristics that underlie the dark personalities lead to an increase in risk-taking activities [15]. In this regard, studies have found strong relationships between the Dark Triad traits and the propensity to take financial risks [16], attitudes toward risky driving [17], health-risk activities, such as substance use and sociosexuality [18,19,20], problematic media use [21], and other deviant behaviours, including aggression [22,23,24], bullying and cyberbullying [25,26,27], and crime [18,28]. Specifically, some differences emerged as regards the influence of Machiavellianism, psychopathy, and narcissism on risky behaviours. Studies evidenced that both psychopathy and narcissism were the most deleterious dark factors regarding risk-taking activities [15,16,19,29]; however, some differences emerged between them in the pattern of risk. Some authors point to psychopathy as the most socially undesirable trait, more strongly linked to behavioural dysregulation and proactive aggression than the other two [1,30], whilst narcissism would be more strongly linked to ego-threat and reactive aggression [18,31]. Machiavellianism has shown an inconsistent pattern of relationships with risk-taking activities. Whereas some studies showed that Machiavellianism was the factor most strongly associated with certain risky behaviours, such as aggression [32], others did not find significant relationships with risk-taking activities [15,29]. Nevertheless, Furnham et al. [9] indicated that each of the relations between the constructs or traits ascertained with the three Dark Triad factors and risky behaviours may differ depending on the type of risky behaviour. From a person-centred approach, we would like to add that not only the relationships but also the composites of levels of scores on each of the factors may differ in accordance with the type of risky behaviour that we try to explain or better understand. However, most of the studies conducted and published hitherto focus on one type of risky behaviour or compare the Dark traits to other personality traits, such as impulsivity or sensation seeking, to shed light on the relationship between the Dark Triad and risky behaviours [15,19]. To better understand individual differences as regards the Dark Triad and what these differences mean for risk behaviour, we need to design and conduct studies that include different types of risky behaviour. 

### 1.2. Individual Profiles from a Person-Centred Approach

The vast majority of studies that have been conducted to analyse the correlates and outcomes of the Dark Triad traits used a variable-centred approach, which allows the analysis of the associations between constructs but neglects the heterogeneity of the population [33]. From a person-centred approach, the existence of different subgroups of individuals is assumed, characterized by a set of patterns of risk that are clustered according to distinct factors or domains [34,35]. Thus, each subgroup might show a differential pattern of associations with distinct risky behaviours [33]. As some authors have pointed out, a variable-centred approach may be desirable when personality traits are considered as continuous, whereas a person-centred approach is more appropriate under the assumption of clinical disorders, i.e., from the perspective that particular compositions based on scores on a set of factors or traits may be seen as a set of latent categories with clinical relevance; a so-called taxonic perspective [13]. Given the subclinical nature of the Dark Triad, a person-centred approach may provide valuable understanding about risk profiles more focused on a person rather than on a trait [11]. 

The existing research on Dark Triad traits from a person-centred approach is scarce; however, a few studies have attempted to identify different subgroups of individuals. Specifically, Chabrol et al. [10] used a sample of high school students to identify four groups based on their scores in the Dark Tetrad (i.e., Machiavellianism, psychopathy, narcissism, and sadism): a low-traits group, a sadistic–Machiavellian group, a psychopathic–narcissistic group, and a high-traits group. The latter was called the Dark Tetrad group. These authors found that the Dark Tetrad group scored higher on antisocial and suicidal behaviour, but some differences emerged between groups as regards depressive symptomatology. On the other hand, García and MacDonald [12] identified three dark personality profiles based on the Dark Triad, namely high malevolent, intermediate malevolent, and low malevolent. This pattern resembles to some extent that found by Kam and Zhou [11], who identified two solutions that obtained similar fit indices and were largely parallel in Dark Triad scores. Specifically, they identified a three-profile solution (i.e., high, middle, low) and a four-profile solution (i.e., high, middle, middle-low, low), which mainly differed quantitatively but not qualitatively in Machiavellianism, psychopathy, and narcissism. Despite the high overlap that was found among subgroups [11,13], some differences emerged between profiles as regards risky behaviours, pointing to the value of both variable-centred and person-centred approaches in terms of external validity and the explanation of risky behaviour [13].

### 1.3. The Current Study

The current study aims to identify different subgroups of individuals on the basis of their scores on the Dark Triad traits (i.e., Machiavellianism, psychopathy, and narcissism) and to analyse the differences between them in a set of risky behaviours (i.e., frequency of substance use, reactive and proactive aggression, risk perception and risk engagement, and problematic internet use). The current study is meant to extend the body of research on the Dark Triad with an approach that takes the heterogeneity of the population as regards the level of Machiavellianism, psychopathy, and narcissism into consideration. Furthermore, this study is meant to advance research by verifying the distinctiveness of the profiles concerning different types of risky behaviours and how they differ according to the pattern of risk. Following the results obtained in previous studies, three hypotheses are proposed. First, at least three subgroups of individuals based on the Dark Triad traits are expected: low, moderate, and high Dark Triad subgroups [11,12]. However, specific profiles may stand out for some specific trait/s of Machiavellianism, psychopathy and/or narcissism [10]. Second, some differences between groups depending on the type of risky behaviour are hypothesised with high Dark Triad subgroups showing the greatest scores in risk-taking. Third, a differential pattern of associations between subgroups and risky behaviour is expected depending on the Dark Triad profiles. On the one hand, stronger relationships are expected between psychopathic and narcissistic subgroups and risk-taking [15,16,19,29]. On the other hand, positive associations between high Dark Triad profiles with risk-engagement and negative associations with risk-perception are expected [15,16].

## 2. Materials and Methods 

### 2.1. Participants and Procedure

The initial sample was composed of 326 participants (47% male), aged 18–34 (*M* = 20.55; *SD* = 1.89), recruited through a combination of non-probabilistic sampling, including both convenience and snowball procedures. This type of participant selection has some advantages, such as accessibility and cost reduction, and is useful when the population of interest is difficult to define. Given the goal of the current study is to explore the existence of different Dark Triad profiles, non-probabilistic sampling methods are an efficient means to achieve this. However, the results must be cautiously interpreted because non-probabilistic methods prevent the generalization of the results beyond the current sample. Of the total sample, ten participants did not complete the questionnaire and were deleted from the dataset [36], giving rise to a final sample composed of 316 participants (48.7% male), aged 18–34 (*M* = 20.55; *SD* = 1.91). A total of 155 participants (21% males) were undergraduates enrolled in the Introductory course of Psychology in a Spanish university. These participants were asked about potential volunteers to participate in the study through their personal contacts, which resulted in the inclusion of 161 more participants (68% males). Of these, 89.4% were students involved in high-level education (75.6% enrolled in different degrees at the university, 18.1% enrolled in vocational training, and 6.3% in other courses). Some gender differences were found between both subsamples (χ^2^ = 44.295, *p* < 0.001), likely because of the female overrepresentation in social sciences [37], whereas no significant differences were found as regards age (*F* = 0.264, *p* = 0.608). All procedures were approved by the Bioethics Committee of the University. Participation was voluntary and only individuals who consented to take part in the study were included as participants. Confidentiality and anonymity were ensured following the legal and ethical standards throughout the investigation process.

### 2.2. Measurements

Dark Triad traits: The Spanish version of the Dark Triad Dirty Dozen was used for the assessment of the Dark Triad traits [32]. This scale is composed of 12 items for the assessment of the three Dark Triad factors (4 items each): Machiavellianism (e.g., “I tend to manipulate others to get my way”, α = 0.85), psychopathy (e.g., “I tend to lack remorse”; α = 0.73), and narcissism (e.g., “I tend to want others to admire me”; α = 0.87), using a 7-point scale ranging from 1 (strongly disagree) to 7 (strongly agree). The Dirty Dozen has demonstrated good construct and convergent validity in previous studies in the Spanish context [32,38]. The internal consistency for the global Dark Triad scale was 0.84.

Substance use: The frequency of use of alcohol, tobacco, and cannabis during the last month was measured by means of three items—one item per substance—(e.g., “How many times have you taken alcoholic drinks in the last month?”), rated on a 6-point scale: 0 (never), 1 (1–2 times), 2 (3–5 times), 3 (6–10 times), 4 (11–20 times), and 5 (more than 20 times).

Reactive and proactive aggression: Aggression was assessed employing the Reactive and Proactive Aggression Questionnaire (RPQ) [39]. This scale is composed of 23 items rated on a 3-point scale from 0 (never) to 2 (often), which evaluates reactive aggression (11 items, e.g., “I reacted angrily when provoked by others”, α = 0.76) and proactive aggression (12 items, e.g., “I had fights with others to show who was on top”, α = 0.81). The two-factor model has been replicated in previous studies using the Spanish RPQ [40]. The internal consistency for the global scale was 0.86.

Risk-perception and risk-engagement: Both risk-perception and risk-engagement were assessed by the Revised Domain-Specific Risk-Taking scale (DOSPERT) that was developed by Blais and Weber (Columbia University, New York, NY, USA) [41] and used previously in the Spanish context [42]. This scale is composed of 30 items that evaluate risky behaviours in five different domains: ethical (e.g., “Having an affair with a married man/woman”), financial (e.g., “Betting a day’s income at a high-stakes poker game”), health/safety (e.g., “Drinking heavily at a social function”), social (e.g., “Disagreeing with an authority figure on a major issue”), and recreational (e.g., “Going down a ski run that is beyond your ability”). The scale was assessed using a 7-point rating scale ranging from 1 (extremely unlikely) to 7 (extremely likely). Both risk-perception and risk-engagement refer to the same 30 items but the instructions for filling in the items slightly differ. Whereas risk-taking was considered as the probability of likelihood of engaging in these risky behaviours, risk-perception was considered as the gut-level assessment of the riskiness of these behaviours, more focused on how risky the participants perceived each situation. The Spanish version of the DOSPERT scale demonstrated good construct and convergent validity regarding both risk perception and risk engagement [43]. The internal consistency for risk-perception was 0.89 and for risk-engagement was 0.87.

Problematic internet use: The Spanish version of the Internet addiction test [44] was used for the assessment of problematic internet use. This scale is composed of 20 items rated on a 5-point scale ranging from 1 (never or very rarely) to 5 (always). The scale previously demonstrated two factors, namely emotional investment (e.g., “How often do you feel depressed, nervous or tense when you are not connected and then these feelings disappear when you connect?”) and performance and time management (e.g., “How often are you connected to the Internet for longer than you had intended?”). This scale has shown adequate construct validity and supported the two-factor structure in the Spanish context [44]. In the current study, the global score was used. The internal consistency for the global scale was 0.91.

### 2.3. Data Analysis

Descriptive statistics and partial correlations among all the study variables controlling for age and gender were analysed using the software package SPSS 25 (IBM, Armonk, NY, USA). Subsequently, a series of latent class analysis (LCA) for continuous variables (i.e., latent profile analysis, LPA) were carried out in Mplus 7.4 (Muthén & Muthén, Los Angeles, CA, USA) [45], including the Dark Triad traits (i.e., Machiavellianism, psychopathy, and narcissism) as class indicators and using the full-information maximum likelihood algorithm for handling missing data. A total of six models were tested (from one to six latent classes). The selection of the best solution was taken after considering both empirical and theoretical criteria [46]. Regarding the former, several fit indices were used to determine the best fitting model: entropy, Bayesian information criteria (BIC), sample size-adjusted BIC (ABIC), and bootstrapped likelihood ratio test (BLRT). Entropy refers to the accuracy in assigning participants to different classes, with values higher than 0.70 and closer to 1 being preferable. On the other hand, lower values of BIC and ABIC (adjusted BIC) indicate a better fit [47]. Furthermore, bootstrapped likelihood ratio test (BLRT) is used to analyse the improvement of a model (k) compared to a one-less class model (k − 1), with low significance values (*p* < 0.05) showing a better fit. Besides, theoretical criteria were used for the selection of the best solution. Specifically, if two models obtained similar solutions, the more parsimonious one was selected if it was in line with the theoretical background. On the other hand, differences between subgroups as regards class indicators were analysed through a set of ANOVAs using SPSS 25, using the Tukey–Kramer index for post hoc comparisons. Furthermore, gender differences and comparisons on risky behaviours were conducted using the Bolck-Croon-Hagenaar (BCH) method [48] in Mplus 7.4, which is the most recommended approach for the examination of the associations between profiles and continuous distal outcomes [49]. 

## 3. Results

### 3.1. Descriptive Statistics and Intercorrelations

Descriptive statistics and partial correlations controlling for age and gender are presented in Table 1. The Bonferroni correction was performed due to the multiple correlations assessed and threshold levels of significance were settled at 0.001 (55 comparisons). The results showed moderate positive correlations of Machiavellianism with both psychopathy and narcissism but weak positive correlations between psychopathy and narcissism. Overall, all the Dark Triad traits were significantly and positively related to reactive and proactive aggression. Specifically, Machiavellianism and psychopathy were positively related to risk engagement and negatively related to risk perception, whereas Machiavellianism and narcissism were positively associated with problematic internet use. No significant correlations were found between the Dark Triad traits and frequency of alcohol, tobacco, and cannabis use.

### 3.2. LPA Including Machiavellianism, Psychopathy, and Narcissism as Indicators

The fit indices for the LPA models are displayed in Table 2. The analysis of the fit indices favoured the five-class model over the four-class, three-class, two-class, and one-class models, respectively. The six-class model outperformed the five-class model on two indices (i.e., entropy and ABIC); however, the BLRT did not show a significant increase over the five-class model, therefore the five-class model was considered as more parsimonious and was selected as the best solution. The analysis of profiles evidenced five different subgroups of individuals that were identified based on their scores on Machiavellianism, psychopathy and narcissism: (1) a narcissistic group (32.4%), (2) a low-Dark Triad group (28.3%), (3) a Machiavellian/narcissistic group (17.7%), (4) a psychopathic group (15.9%), and (5) a Machiavellian/psychopathic group (5.6%). The distribution of the groups is presented in Figure 1. The most prevalent group was the narcissistic group, which obtained lower scores in Machiavellianism and psychopathy and the second-highest score in narcissism. The low-Dark Triad group was the second-largest subgroup and obtained the lowest scores in all the Dark Triad factors. The Machiavellian/narcissistic group scored higher in Machiavellianism and narcissism but moderate in psychopathy, whereas the psychopathic group showed the opposite trend; that is, higher scores in psychopathy along with lower scores in Machiavellianism and narcissism. The smallest group was the Machiavellian/psychopathic, which scored higher in Machiavellianism and psychopathy and lower in narcissism. Descriptive statistics and comparisons between subgroups in Dark Triad class indicators are presented in Table 3. The results evidenced significant differences among subgroups in all the Dark Triad traits. Specifically, the Tukey-Kramer index for post hoc comparisons indicated significant differences in Machiavellianism between subgroups except for the narcissistic and psychopathic/narcissistic, which did not differ in the levels of Machiavellianism. Regarding psychopathy, post hoc comparisons showed significant differences among subgroups except for between low-Dark Triad and narcissistic and between psychopathic and Machiavellian/psychopathic. Finally, the results evidenced significant differences between subgroups in narcissism, except for between the Machiavellian/psychopathic with the narcissistic, Machiavellian/narcissistic, and psychopathic.

Significant differences were found in the distribution of subgroups regarding gender (χ^2^ = 20.165, df = 4, *p* < 0.001). Specifically, the Machiavellian/psychopathic group included significantly more males in comparison with the narcissistic group (χ^2^ = 12.060, *p* < 0.001), the low-Dark Triad group (χ^2^ = 11.291, *p* < 0.001), and the Machiavellian/narcissistic group (χ^2^ = 4.254, *p* < 0.05). In a similar vein, the narcissistic group (χ^2^ = 6.595, *p* < 0.01) and the low-Dark Triad group (χ^2^ = 6.898, *p* < 0.01) included significantly more females compared to the psychopathic group.

### 3.3. Differences between Subgroups in Risky Behaviours

Comparisons in risky behaviours for the five-class solution are displayed in Table 4. Overall, significant differences were found between subgroups as regards reactive and proactive aggression, risk perception and risk engagement, and problematic internet use. However, no significant differences were found regarding the frequency of substance use, including alcohol, tobacco, and cannabis. Specifically, the BCH results evidenced a differential pattern of association between subgroups. The Machiavellian/narcissistic obtained the highest scores in reactive aggression and risk engagement, whereas the Machiavellian/psychopathic scored higher in proactive aggression and internet problematic use. Machiavellian/narcissistic group significantly differed from the high-narcissistic group in risk engagement, and reactive and proactive aggression and from the psychopathic group in risk engagement, reactive and proactive aggression, and problematic internet use. In addition, the psychopathic group only differed from the Machiavellian/psychopathic group in problematic internet use, but no significant differences emerged with the narcissistic group in this type of behaviour. Likewise, the results showed significant differences between the Machiavellian/psychopathic and the narcissistic group in risk perception and proactive aggression. Finally, the low-Dark Triad group showed the highest score in risk perception as well as the lowest scores in risk engagement, reactive and proactive aggression, and problematic internet use. However, these differences were statistically significant in comparison with all the subgroups in risk engagement and proactive aggression.

## 4. Discussion

The study of Dark Triad traits has been widely considered from a variable-centred approach. Only a handful of studies have previously focused on the associations of Machiavellianism, psychopathy, and narcissism with risk-taking activities using a person-centred approach. The current study aimed to identify different subgroups of individuals based on the Dark Triad traits and delve into the differences among them in a set of risky behaviours. Overall, five profiles were identified, namely low-Dark Triad, narcissistic, Machiavellian/narcissistic, psychopathic, and Machiavellian/psychopathic. No significant differences were found as regards the frequency of substance use, but some differences between subgroups were evidenced in reactive and proactive aggression, risk perception, risk engagement, and problematic internet use. Specifically, the Machiavellian/narcissistic and the Machiavellian/psychopathic profiles scored higher in risky behaviours than the other subgroups, whereas the low-Dark Triad evidenced higher scores in risk perception. On the other hand, some gender differences in the composition of the groups were found. Specifically, the low-Dark Triad and the narcissistic subgroups included more females, whereas the Machiavellian/narcissistic, psychopathic, and Machiavellian/psychopathic groups were predominantly male clusters. This is in line with previous findings, which showed higher scores in all Dark Triad traits in males compared to females [23,32,38], resulting in a high proportion of men in the high-Dark Triad groups. Furthermore, even when men have proven to be more narcissistic than women [50], in the current study this did not lead to a “merely or predominantly male” narcissistic profile, but to different profiles combining Dark Triad traits. Nevertheless, differences in the composition of the subgroups should be cautiously interpreted given the characteristics of the sample, which may overestimate the similarity between males and females in the community population.

The results partially support the first hypothesis, which stated that at least three subgroups would be identified (i.e., low, moderate, and high), but other profiles might emerge according to the Dark Triad traits. In this regard, the three basic profiles were not identified but five different subgroups, which differ not only quantitatively but also qualitatively on the Dark Triad traits (i.e., low-Dark Triad, narcissistic, Machiavellian/narcissistic, psychopathic, and Machiavellian/psychopathic). These findings indicate that individuals may be clustered in different profiles according to their levels of Machiavellianism, psychopathy, and narcissism beyond the traditional distinction of low, moderate, and high risk [51], which supports the usefulness of the person-centred approach in the study of the Dark Triad traits. Nevertheless, more studies are needed to replicate and validate these profiles, since different classifications may arise based on the sample, measures used, or other factors. Certainly, the results of the current study differ from those found in previous studies, which showed a better fit for a one-class solution [13], and the traditional classification of low, moderate, and high [11,12]. The high overlap that was found between subgroups in these studies and the lack of qualitative differences among them failed to demonstrate the superiority of the person-centred approach over the variable-centred approach [12,13]. However, and concurring with Chabrol et al. [10], the results of the current study support the distinction of different profiles based on their scores on the Dark Triad that differ not only quantitatively but also qualitatively. This fact points to the possibility that some individuals may score higher than others only in one or more dark factors, which can determine to some extent the relation with specific outcomes. Nevertheless, the expected high-Dark Triad subgroup—that is, a subgroup characterized by higher scores in all the Dark Triad traits—was not identified. The nature of the sample used in this study may partly explain this result, since most participants were undergraduates or high-level education participants, who may score very different in terms of dark personality traits and risky behaviours from forensic, clinical, or even community samples and other developmental periods [9,52,53]. It is noteworthy that the identified profiles differ from those found in previous studies, which used adult samples, both undergraduates and other community samples [11,12,13], but are in line with the only study that analysed dark subgroups in high-school students [10]. Future studies should aim to replicate these findings across different developmental stages considering the role of specific variables, such as gender or socioeconomic status, in the identification of subgroups. 

As the second hypothesis stated, the results suggest a strong association between the Dark Triad and risky behaviours [15,16,29]. Despite the differences that were found among subgroups, the current results suggest that individuals encompassed in each of the four high-Dark Triad profiles (i.e., narcissistic, Machiavellian/narcissistic, psychopathic, and Machiavellian/psychopathic) engage in more risky behaviours than those individuals who score lower in the Dark Triad traits. One unexpected result is the lack of significant relationships between the Dark Triad traits and the frequency of substance use, including alcohol, tobacco, and cannabis, both from variable-centred and person-centred approaches. This result is not in line with previous findings that showed strong relationships between the Dark Triad and health-risk activities, including substance use [19,20]. From an evolutionary model of risky behaviour, some high-risk activities such as substance use may be common among young people [54]. Hence, the underlying mechanism to detect and respond to costs and benefits might lead to the consideration of substance use as normative behaviour in this sample, which reduces the weight of individual differences [54]. Another possible explanation is that the Dark Triad does not exert a direct effect on substance use but an indirect effect through specific mechanisms of approach and avoidance behaviours, such as inhibition and activation systems [20,55], and antagonism [19], which may also differ between Dark Triad traits. In this line, Flexon, Meldrum, Young, and Lehmann [56] found that the Dark Triad no longer predicted substance use after controlling for the effect of self-control.

These findings partially support the third hypothesis, since a differential pattern of associations between subgroups was found, but, despite the fact that the Dark Triad is positively associated with risk engagement and negatively associated with risk perception, psychopathic and narcissistic profiles do not show the strongest relationship with risky behaviour. Overall, significant differences between subgroups emerged as regards risk perception, risk engagement, reactive and proactive aggression, and problematic internet use. Specifically, the high-Machiavellian/narcissistic and the high-Machiavellian/psychopathic subgroups display the highest levels of risky behaviour, whilst the low-Dark Triad score higher in risk perception. Given the significant correlations that were found between Dark Triad and risk perception as well as between risk perception and risk engagement, it is likely that the perception of risk plays a significant role in the relationship between Dark Triad and risk behaviours [57]. However, the current analyses do not allow a mediation hypothesis to be tested. Future studies should be aimed at disentangling the potential mediating role of risk perception on the relation between the Dark Triad traits and risk-taking. On the other hand, the trend in the behavioural patterns seems slightly different among subgroups. Whereas the Machiavellian/narcissistic profile is strongly related to risk engagement and reactive aggression, the Machiavellian/psychopathic group obtained the highest scores in proactive aggression and problematic internet use as well as the lowest scores in risk perception. These findings also highlight the relevance of using a person-centred approach for the study of the Dark Triad traits and risk-taking activities. At a correlational level, Machiavellianism, psychopathy, and narcissism were significantly associated with all these behaviours, except for psychopathy with problematic internet use and narcissism with risk perception and risk engagement. However, the results of the LPA evidence that three Dark Triad traits are neither independently nor similarly related to these behaviours, but the combination of specific traits may constitute more socially undesirable profiles than each of the Dark Triad traits separately. 

These results are noteworthy due to several reasons. Firstly, two “pure” subgroups were identified as regards psychopathy and narcissism though not in the case of Machiavellianism. This is in line with previous studies that found support for the independent contribution of each of the Dark Triad, which may reflect distinct dark personalities [1,5,58]. Secondly, contrary to previous findings, psychopathy and narcissism are not the most aversive Dark Triad traits in terms of risk behaviour [18,30], at least in this specific sample. However, it should be noted that a subgroup high in psychopathy and narcissism was not identified in the current study, which makes it difficult to determine whether the combination of the two characteristics could have a stronger relationship with risky behaviour than other combinations. Thirdly, it seems that the dark personality characteristics have a stronger relationship with risky behaviours depending on the presence of Machiavellianism. Contrary to what some authors have suggested [15,29], Machiavellianism seems a key factor in the explanation of risk-taking activities. The profiles characterized by Machiavellianism/narcissism and Machiavellianism/psychopathy evidence higher levels of risky behaviour in comparison with the “pure” narcissistic and psychopathic profiles. Hence, the results of the current study suggest that the combination of the Dark Triad may better explain the involvement in risky behaviours rather than the consideration of each of its constituents separately. This is also in line with previous studies that found more negative outcomes in those individuals who were classified in the high Dark-Triad profiles [9,10,12]. Thus, even though there may be “pure” Machiavellians, psychopaths, and narcissistic profiles, it seems that the combination of the Dark Triad traits leads to more negative outcomes regarding risky behaviour. Therefore, using a person-centred approach might provide valuable insights in this regard.

Notwithstanding the prior contributions, the current study should be interpreted in line with the following limitations. First, the sample of analysis mostly include participants involved in the high-level education, which may significantly differ in terms of personality and risky behaviour not only from forensic samples but also from other individuals in the community. Furthermore, non-probabilistic sampling methods were used for participant selection. This limits the representativeness of the sample and, consequently, the generalization of the results to the entire population. Given the subclinical nature of the Dark Triad, future studies must include larger sample sizes and use probabilistic research designs that allow for generalization of the results to the community population. Secondly, the LPA was conducted using the Dirty Dozen measure of the Dark Triad. Despite the fact that the Dirty Dozen has previously shown good psychometric properties [23,32], some authors have argued that this scale does not capture all the characteristics encompassed in the factors of Machiavellianism, psychopathy, and narcissism [58,59]. Hence, future research should replicate the current findings by using other measures of the Dark Triad, such as the Short Dark Triad (SD3) [59], to test the validity of the profiles. Furthermore, the strong correlations found between Machiavellianism, psychopathy, and narcissism may cause an overlap in the profiles that should be taken into consideration when interpreting the results. Thirdly, data used in this study were mainly collected through self-report questionnaires; therefore, results might be partially influenced by shared method variance [60]. The use of different sources of information, as well as different methods of data collection, must be considered in future studies. Finally, the cross-sectional nature of this study prevents the consideration of dynamic profiles across different developmental periods. Longitudinal studies are needed to assess the potential stability of the profiles over time.

## 5. Conclusions

In conclusion, the current findings suggest that different profiles based on the Dark Triad traits may be identified, that differ not only quantitatively but also qualitatively. Specifically, five subgroups were identified based on the scores on Machiavellianism, psychopathy, and narcissism, namely low-Dark Triad, narcissistic, Machiavellian/narcissistic, psychopathic, and Machiavellian/psychopathic. Furthermore, these results suggest that the study of dark personalities from a person-centred approach contributes to the comprehension of the personality profiles associated with risk-taking behaviours. Overall, the results evidenced that the combination of Dark Triad traits leads to more negative outcomes in terms of risky behaviour compared to the “pure” psychopathic and narcissistic profiles and the low-Dark Triad. However, different trends in the patterns of risk were found among subgroups. This study highlights the consideration of the specific facets of Machiavellianism, psychopathy, and narcissism simultaneously to better understand the involvement in risky behaviours. 

## Figures and Tables

**Figure 1 ijerph-17-06194-f001:**
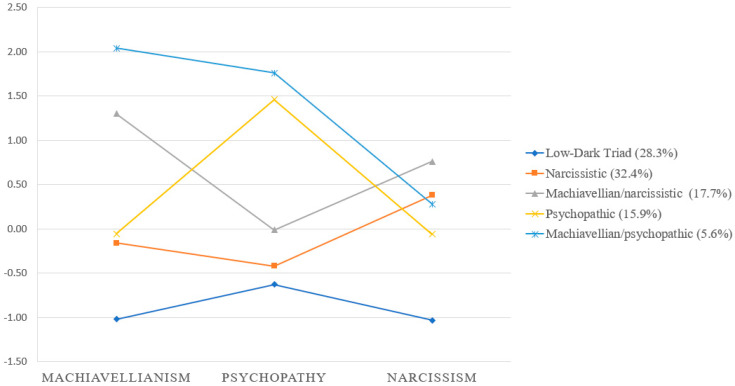
Mean Z-scores for the five-class solutions including the Dark Triad traits as class indicators.

**Table 1 ijerph-17-06194-t001:** Descriptive statistics and partial correlations among all the study variables controlling for gender and age.

	1	2	3	4	5	6	7	8	9	10	11
1. Machiavellianism	−										
2. Psychopathy	0.41 *	−									
3. Narcissism	0.40 *	0.13	−								
4. Alcohol frequency	0.15	0.10	−0.01	−							
5. Tobacco frequency	0.16	0.11	−0.09	0.31 *	−						
6. Cannabis frequency	0.15	0.13	−0.06	0.28 *	0.51 *	−					
7. Reactive aggression	0.36 *	0.20 *	0.29*	0.14	0.14	0.06	−				
8. Proactive aggression	0.48 *	0.33 *	0.29*	0.28 *	0.11	0.05	0.60 *	−			
9. Risk perception	−0.30 *	−0.26 *	−0.05	−0.31 *	−0.20 *	−0.16	−0.23 *	−0.27 *	−		
10. Risk engagement	0.40 *	0.22 *	0.15	0.44 *	0.20 *	0.26 *	0.31 *	0.42 *	−0.46 *	−	
11. Internet use	0.34 *	0.16	0.23*	0.04	−0.05	−0.02	0.27 *	0.27 *	−0.10	0.26 *	-
*M* *(SD)*	10.70 (4.86)	9.79 (4.64)	14.07 (5.67)	2.08 (1.46)	1.24 (2.01)	0.66 (1.40)	20.03 (3.82)	15.29 (3.13)	91.98 (15.27)	80.49 (16.31)	34.67 (11.23)
Range	4–26	4–28	4–28	0–5	0–5	0–5	12–32	12–29	49–130	36–129	20–84

*Note.* Internet use = problematic internet use. * Significant *p* value after applying the Bonferroni correction (*p* < 0.001).

**Table 2 ijerph-17-06194-t002:** Fit indices for latent profile models including Machiavellianism, psychopathy, and narcissism as class indicators.

	Entropy	BIC	ABIC	BLRT (*p*)
Class 1		5384.49	5365.46	
Class 2	0.646	5302.89	5271.17	104.432 ***
Class 3	0.711	5272.72	5228.32	52.99 ***
Class 4	0.788	5271.96	5214.87	23.59 ***
**Class 5**	**0.764**	**5257.81**	**5188.04**	**36.974 *****
Class 6	0.781	5269.29	5186.83	11.35

*Note*. BIC = Bayesian information criterion; ABIC = adjusted BIC; BLRT = bootstrapped likelihood ratio test. Bold-faced values represent the best-fit solution. *** *p* < 0.001.

**Table 3 ijerph-17-06194-t003:** Descriptive statistics and differences among subgroups in Dark Triad traits class indicators.

	Low-Dark Triad (*n* = 82; 61% Females)	Narcissistic (*n* = 101; 60.6% Females)	Machiavellian/ Narcissistic (*n* = 55; 47.3% Females)	Psychopathic (*n* = 48; 37.5% Females)	Machiavellian/ Psychopathic (*n* = 15; 20% Females)	*F*	Partial η^2^
	*M* (*SD*)	*M* (*SD*)	*M* (*SD*)	*M* (*SD*)	*M* (*SD*)		
Machiavellianism	5.73 (1.47) _a_	9.66 (1.96) _b_	16.33 (2.02) _d_	10.15 (2.34) _b_	19.73 (2.49) _e_	333.874 ***	0.82
Psychopathy	6.93 (2.37) _a_	7.76 (2.38) _a_	9.42 (2.59) _c_	15.39 (2.32) _d_	16.60 (2.35) _d_	141.243 ***	0.65
Narcissism	8.32 (3.05) _a_	16.16 (4.29) _d_	18.28 (4.61) _e_	13.73 (4.85) _b_	15.60 (4.32) _b,d,e_	60.321 ***	0.45

*Note*. η^2^ = eta square effect size. Means with different subscripts (a, b, c, d, e) were significantly different (*p* < 0.05) in post hoc pairwise comparisons (subscript a represents the lowest score/s in the analysed indicator). *** *p* < 0.001.

**Table 4 ijerph-17-06194-t004:** Comparisons across subgroups on risky behaviours using the BCH method.

	Low-Dark Triad	Narcissistic	Machiavellian/Narcissistic	Psychopathic	Machiavellian/Psychopathic	χ^2^
	*M* (*SE*)	*M* (*SE*)	*M* (*SE*)	*M* (*SE*)	*M* (*SE*)	
Alcohol frequency	1.72 (0.17) _a_	2.07 (0.18) _a_	2.32 (0.23) _a_	2.43 (0.27) _a_	2.12 (0.45) _a_	7.275
Tobacco frequency	0.99 (0.23) _a_	0.89 (0.24) _a_	1.98 (0.37) _a_	1.73 (0.38) _a_	0.76 (0.43) _a_	9.012
Cannabis frequency	0.39 (0.13) _a_	0.42 (0.15) _a_	1.14 (0.29) _a_	0.88 (0.27) _a_	0.92 (0.53) _a_	8.928
Reactive aggression	18.37 (0.42) _a_	19.52 (0.43) _a,b_	22.97 (0.59) _c_	20.61 (0.55) _b,d_	20.52 (1.13) _a,b,c,d_	45.884 ***
Proactive aggression	13.47 (0.20) _a_	14.76 (0.29) _b_	17.42 (0.63) _d_	15.58 (0.54) _b,c_	17.65 (0.93) _c,d_	69.788 ***
Risk perception	96.99 (2.02) _d_	93.43 (1.79) _b,c,d_	88.64 (2.12) _a,b_	90.20 (2.51) _a,b,c_	80.16 (5.03) _a_	17.618 ***
Risk engagement	71.01 (1.84) _a_	81.33 (2.04) _c,d_	90.76 (2.49) _e_	79.64 (2.38) _b,c_	86.03 (4.96) _b,d,e_	46.617 ***
Internet use	29.32 (1.04) _a_	36.37 (1.37) _b,c_	38.39 (2.24) _c,d_	32.29 (1.40) _a,b_	41.71 (3.13) _c,d_	32.366 ***

*Note.* Internet use = problematic internet use. Means with different subscripts (a, b, c, d, e) were significantly different (*p* < 0.05) in post hoc pairwise comparisons (subscript a represents the lowest score/s in the analysed indicator). *** *p* < 0.001.

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
