# Peer review of "Dark Triad Traits and Risky Behaviours: Identifying Risk Profiles from a Person-Centred Approach"

_ijerph, 2020, doi:10.3390/ijerph17176194_

Round 1

Reviewer 1 Report

The topic on the connection between dark triad traits and health behaviors is very interesting and quite new to this reviewer. However, this study employed non-random sampling methods (convenience and snowball sampling) and does not deal with any potential sampling errors or selection bias. Thus, the authors need to justify this non-random sampling methods in research design and how potential errors could be controlled should be explained in order to make the current statistical results robust and reliable.

Furthermore, the target group of the study is very purposive; Why do we need to conduct this study with university students? The authors need to explain whether the results will be same with he other age groups or the same age group who are not attending universities.

Lastly, there is no control variable (e.g., SES) in the analysis. There are lots of potential confounding third factors affecting the relationships between dark triad traits and negative health behaviors. They should be identified properly and controlled for in the analysis. 

Reviewer 2 Report

The term "Dark" appears in the text in quotes and without it, capitalized and not capitalized. Is it scientific? Please, unify the notation.
Please, indicate what do you mean by "the “dark” personalities". Is it a common term? I believe that one should be very careful with definitions like this.
Some proofreading should be performed. Authors tend to mix American and British English rules. Some articles are missing throughout the text.
Were all the participants students in your study? This can significantly influence this research due to the positive selection effect. The sample can be not representative enough and the results cannot be expanded to the whole society.

How the frequency of use of alcohol, tobacco, and cannabis is connected with being a "Dark personality"? Many "bright" and "light" personalities used these, and many people do not misbehave when doing it.

The estimation of risk engagement should involve independent evaluation of risk taken, otherwise, it can be influenced (or falsified?) by respondents. Young people tend to overestimate or underestimate risks.

In my country disagreeing with an authority figure on a major issue is not a big risk inside academic society.

Table 1 will be more representative as a diagram. It is hard for a reader to compare numbers now.

Figure 1 is barely explained and discussed. It is just a representation of the obtained data.

You cannot claim that gender differences emerged in such a small and non-representative sample. The similarity between male and female university students is much more than between male student and male dock worker. You should take social groups into account here. I believe that the found "similarities" possibly belong to the social but not gender differences.

In my opinion, the study lacks some intellectual tests or students' marks assessment. Machiavellism requires strong intelligence to properly perform social manipulations. This can explain the existence of a separate "psychopathic \ narcissistic" group.

Current conclusions are weak and barely grounded by the obtained results.

Nevertheless, I like this study and can recommend it for publication after some improvements.

Round 2

Reviewer 1 Report

The authors have tried to incorporate this reviewer's suggestions and comments as much as possible.

Reviewer 2 Report

Dear authors!
Thank you for providing a revised version of your manuscript.
I greatly value the efforts you put in improving the manuscript and understand that the experimental part of the study cannot be significantly changed in such a short revision time. I highly recommend you to take into account the social differences of respondents in your further studies because the social being determines consciousness more than natural abilities or gender.
I will recommend the paper for publication and wish all the best to the authors.